# Long-Term Human Immune Reconstitution, T-Cell Development, and Immune Reactivity in Mice Lacking the Murine Major Histocompatibility Complex: Validation with Cellular and Gene Expression Profiles

**DOI:** 10.3390/cells13201686

**Published:** 2024-10-12

**Authors:** Milita Darguzyte, Philipp Antczak, Daniel Bachurski, Patrick Hoelker, Nima Abedpour, Rahil Gholamipoorfard, Hans A. Schlößer, Kerstin Wennhold, Martin Thelen, Maria A. Garcia-Marquez, Johannes Koenig, Andreas Schneider, Tobias Braun, Frank Klawonn, Michael Damrat, Masudur Rahman, Jan-Malte Kleid, Sebastian J. Theobald, Eugen Bauer, Constantin von Kaisenberg, Steven R. Talbot, Leonard D. Shultz, Brian Soper, Renata Stripecke

**Affiliations:** 1Institute for Translational Immune-Oncology, Cancer Research Center Cologne-Essen (CCCE), Faculty of Medicine and University Hospital Cologne, University of Cologne, 50931 Cologne, Germany; milita.darguzyte@uk-koeln.de (M.D.); michael.damrat@uk-koeln.de (M.D.); masudur.rahman@uk-koeln.de (M.R.); jan-malte.kleid@uk-koeln.de (J.-M.K.); 2Department I of Internal Medicine, Center for Integrated Oncology Aachen Bonn Cologne Düsseldorf, Center for Molecular Medicine Cologne (CMMC), Faculty of Medicine and University Hospital Cologne, University of Cologne, 50931 Cologne, Germany; daniel.bachurski@uni-koeln.de (D.B.); patrick.hoelker@uk-koeln.de (P.H.); nima.abedpour@uni-koeln.de (N.A.); rahil.gholamipoorfard@uni-koeln.de (R.G.); sebastian.theobald@uk-koeln.de (S.J.T.); 3Department II of Internal Medicine, Center for Molecular Medicine Cologne (CMMC), Faculty of Medicine and University Hospital Cologne, University of Cologne, 50931 Cologne, Germany; pantczak@uni-koeln.de; 4Cologne Cluster of Excellence on Cellular Stress Responses in Ageing-Associated Diseases, 50931 Cologne, Germany; 5Mildred Scheel School of Oncology Aachen Bonn Cologne Düsseldorf, Faculty of Medicine and University Hospital of Cologne, University of Cologne, 50931 Cologne, Germany; 6Department of Translational Genomics, Cancer Research Center Cologne-Essen (CCCE), Faculty of Medicine and University Hospital Cologne, University of Cologne, 50931 Cologne, Germany; 7Center for Molecular Medicine Cologne (CMMC), Faculty of Medicine and University Hospital Cologne, University of Cologne, 50931 Cologne, Germany; hans.schloesser@uk-koeln.de (H.A.S.); kerstin.wennhold@uk-koeln.de (K.W.); martin.thelen@uk-koeln.de (M.T.); maria.garcia-marquez@uk-koeln.de (M.A.G.-M.); 8Department of General, Visceral, Cancer and Transplantation Surgery, Faculty of Medicine and University Hospital Cologne, University of Cologne, 50931 Cologne, Germany; 9Department of Hematology, Oncology, Hemostasis and Stem Cell Transplantation, Hannover Medical School, 30625 Hannover, Germany; koenig.johannes@mh-hannover.de (J.K.); schneider.andreas@mh-hannover.de (A.S.); braun.tobias@mh-hannover.de (T.B.); 10Department of Computer Science, Ostfalia University of Applied Sciences, 38302 Wolfenbuettel, Germany; frank.klawonn@helmholtz-hzi.de; 11Biostatistics Group, Helmholtz Centre for Infection Research, 38124 Braunschweig, Germany; 12Division of Infectious Diseases, Department I of Internal Medicine, University Hospital of Cologne, 50931 Cologne, Germany; 13German Center for Infection Research (DZIF), Partner Site Bonn-Cologne, 50931 Cologne, Germany; 14Institute of Transfusion Medicine, Faculty of Medicine and University Hospital Cologne, University of Cologne, 50931 Cologne, Germany; eugen.bauer@uk-koeln.de; 15Department of Obstetrics, Gynecology and Reproductive Medicine, Hannover Medical School, 30625 Hannover, Germany; vonkaisenberg.constantin@mh-hannover.de; 16Institute for Laboratory Animal Science, Hannover Medical School, 30625 Hannover, Germany; talbot.steven@mh-hannover.de; 17The Jackson Laboratory, Bar Harbor, ME 04609, USA; lenny.shultz@jax.org

**Keywords:** stem cell transplantation, NSG, humanized mice, MHC, HLA, T-cell, IFN, lentivirus, immunization

## Abstract

Background: Humanized mice transplanted with CD34^+^ hematopoietic cells (HPCs) are broadly used to study human immune responses and infections in vivo and for testing therapies pre-clinically. However, until now, it was not clear whether interactions between the mouse major histocompatibility complexes (MHCs) and/or the human leukocyte antigens (HLAs) were necessary for human T-cell development and immune reactivity. Methods: We evaluated the long-term (20-week) human hematopoiesis and human T-cell development in NOD Scid Gamma (NSG) mice lacking the expression of MHC class I and II (NSG-DKO). Triplicate experiments were performed with HPCs obtained from three donors, and humanization was confirmed in the reference strain NOD Rag Gamma (NRG). Further, we tested whether humanized NSG-DKO mice would respond to a lentiviral vector (LV) systemic delivery of HLA-A*02:01, HLA-DRB1*04:01, human GM-CSF/IFN-α, and the human cytomegalovirus gB antigen. Results: Human immune reconstitution was detectable in peripheral blood from 8 to 20 weeks after the transplantation of NSG-DKO. Human single positive CD4^+^ and CD8^+^ T-cells were detectable in lymphatic tissues (thymus, bone marrow, and spleen). LV delivery harnessed the detection of lymphocyte subsets in bone marrow (αβ and γδ T-cells and NK cells) and the expression of HLA-DR. Furthermore, RNA sequencing showed that LV delivery increased the expression of different human reactome pathways, such as defense responses to other organisms and viruses. Conclusions: Human T-cell development and reactivity are independent of the expression of murine MHCs in humanized mice. Therefore, humanized NSG-DKO is a promising new model for studying human immune responses, as it abrogates the xenograft mouse MHC interference.

## 1. Introduction

Major histocompatibility complexes (MHCs) were discovered over a century ago in mice through the elucidation of the roles of genetic determinants of the immune system in both transplantation and cancer development [1]. The discovery of the more genetically diversified human MHC homologue, the human leukocyte antigen (HLA), enabled serological assays for optimizing the matching between donors and recipients of solid organ transplantation [2]. In follow-up, hematopoietic stem cell transplantation (HCT) with bone marrow from HLA-matched donors into patient recipients with leukemia after high-dose radiation/chemotherapy resulted in engraftment, chimerism, and eventual graft-versus-leukemia effects [3]. As a result, HCT is nowadays an established curative clinical procedure for patients with hematologic malignancies or hematologic genetic defects.

Intriguingly, the transplantation of irradiated immune-deficient mice with enriched human hematopoietic stem cells (HSCs) or human progenitor cells (HPCs) results in mouse/human chimeric hematopoiesis and a not fully functional human immune system (HIS) [4]. NSG mice combine the nonobese diabetic (NOD) inbred genetic background with the *Prkdc^scid^* mutation, causing severe combined immunodeficiency, and they harbor a complete null mutation of the common cytokine receptor gamma chain (*Il2rγ^null^*). The transplantation of these mice with purified cord blood (CB)-derived CD34^+^CD38^-^ HSCs achieved high and prolonged human chimerism with the effective development of T-cells displaying MHC-restricted cytotoxic functions [5]. Further, immune-deficient radio-resistant NRG mice (combining the NOD and *IL2rγ^null^* stocks and incorporating targeted mutations in the recombination-activating *gene-1*, *Rag1^null^*) showed comparable humanization with CD34^+^ HPCs but less irradiation-induced tissue damage and xeno-graft-versus-host (xeno-GvHD) disease than NSG mice [6].

Currently, several immune-deficient mouse strains transplanted with CD34^+^ HPCs provide relevant models for the elucidation of the human immune system, infectious diseases, and cancer immune biology, and they are becoming promising platforms for testing human-specific immune therapies [7,8]. Although humanized mice recapitulate several aspects of HCT in humans, there is a scarce and delayed development of human T-cells. The lack or low levels of human cytokines, poor innate immune cell development, and underdeveloped lymphatic structures are some of the issues identified as contributing to the weak development of human T-cells [9].

Some of these problems were ameliorated when we applied human-induced dendritic cells (iDCs) expressing human GM-CSF/IFN-α and antigens to NRG mice humanized with cord blood CD34^+^ cells (huNRG) [10,11]. The administration of iDCs promoted lymphatic regeneration and antigen-specific functional and polyclonal T- and B-cell immune responses against human cytomegalovirus (HCMV) pp65 and gB antigens [10,11]. We hypothesized that the engineered human DCs could supply signals needed for positive and negative human T-cell selections in mouse lymphatic tissues. This promoted the full maturation of T- and B-cells with functional T-cell receptors (TCRs) and B-cell receptors (BCRs), respectively.

Other groups demonstrated that the humanization of transgenic NSG- and NRG-derived strains expressing human class I and class II MHCs/HLAs, as expected, showed improvements in T-cell development and cytotoxic functions [12,13,14,15,16]. Due to the difficulties of generating transgenic mice with several different HLA alleles, in previous work, we explored the lentiviral vector (LV) in vivo delivery of HLA-DRB1*04:01 (DR4) into huNRG mice [17]. Upon the co-administration of LV-DR4 with an additional LV “vaccine” expressing human GM-CSF/IFN-α and gB, we observed the activation of T- and B-cell development [17].

Previous to our work, NSG-DKO lacking the expression of class I and II mouse MHCs were used to engraft mature human T-cells [18]. Brehm et al. reported a more gradual and persistent expansion of exogenously administered adoptive human T-cells and less xeno-GvHD in NSG-DKO mice compared to the parental NSG strain, but eventually, xenograft reactivity occurred for both strains after several weeks [18].

Since the engraftment of NSG-DKO with CD34^+^ HPCs was not previously reported, we evaluated whether this was possible and whether B- and T-cells would develop. Here, we used three different HPC donors to assess human T-cell development in humanized NSG-DKO (huNSG-DKO) mice. Further, we evaluated the effects of the LV-mediated gene delivery of HLA-DR4, HLA-A*02:01 (A2.1), human GM-CSF/IFN-α, and gB in the activation of human immune responses. We performed analyses of human reconstitution, analyzing cell lineages (using multicolor flow cytometry and high-dimensional mass cytometry) and gene expression profiles (using bulk mRNA sequencing). Our data demonstrate that the expression of mouse MHCs was dispensable for T-cell development in huNSG-DKO mice. In addition, combinations of the LV-mediated expression of human HLAs, human cytokines, and a viral antigen in huNSG-DKO mice showed a profound effect on the human and mouse mRNA profiles, indicating the activation of innate and adaptive immune regulatory pathways.

## 2. Materials and Methods

### 2.1. Lentivirus Production

The self-inactivation third-generation lentivirus vectors LV-HLA-DR4/fLuc, LV-HLA-A2.1/fLuc, and LV-hGM-CSF/hIFN-α/HCMV-gB (Appendix A) were produced and titered as previously described [11,17].

### 2.2. HPC Selection

The study was conducted according to the guidelines of the Declaration of Helsinki. The collection of cord blood specimens was approved by the Ethics Committee of the Hannover Medical School (MHH) and was performed by the Research Obstetric Biobank (approval number 1303 to Constantin von Kaisenberg). The use of the cord blood specimens to humanize mice was approved by the Ethics Committee of the MHH (approval number 4837 to Renata Stripecke). The transfer of the cord blood samples from the MHH to the University Hospital of Cologne (UKK) was approved by the UKK Ethics Committee (approval number 22-1423_3 to Renata Stripecke). After CB collection, CD34^+^ cells were positively selected through two consecutive runs using immune magnetic beads kit (Miltenyi Biotec, Bergisch-Gladbach, Germany) and cryopreserved as previously described [10,11]. The HLA genotypes of the cord blood units were obtained through sequencing. In detail, the genotyping was performed with the NGS method via Illumina (San Diego, CA, USA), using an in-house kit accredited by DAkkS (Deutsche Akkreditierungsstelle) and EFI (European Federation for Immunogenetics). This kit contains in-house PCR primers combined with GoTag Long PCR Master Mix provided by Promega (Madison, WI, USA) for initial PCR amplification. The reagents for library preparation were from NEBNext Ultra II DNA Library Prep Kit (Frankfurt am Main, Germany) and were used according to the manufacturer’s recommendation. The index PCR was carried out with the CE-certified NGSgo^®^-IndX Indices & Adapters Illumina Set A + B from GenDx (Utrecht, The Netherlands). The cleaning steps and fragment-size selection were performed using AMPure XP beads with BeckmanCoulter. Before sequencing, the library was quantified on the StepOnePlus qPCR device from AppliedBiosystems using the KAPA Library Quantification Kit from Roche (Basel, Switzerland).. Sequencing was carried out with the Illumina MiSeq (San Diego, CA, USA) on a standard flow cell using the 500 v2 chemistry. The generated sequences were analyzed with NGSengine v2.31.0 and implemented the IMGT 3.50.0 database. Although HLA-DRB1*04:01 and HLA-A*02:01 should be relatively frequent in the Caucasian population [19], we were not able to obtain HLA-DRB1*04:01-positive CB from our donor population to match the LV-HLA-DR4/fLuc delivery. Therefore, we selected three cord blood units that were positive for HLA-A*02:01, which could match the LV-HLA-A2.1/fLuc delivery.

### 2.3. Generation of Humanized Mice

The animal protocols for mouse studies were approved by the Lower Saxony Office for Consumer Protection and Food Safety–LAVES (approval number 33.19-42502-04-19/3336 to Renata Stripecke) and performed according to the German animal welfare act and EU directive 2010/63. The well-being of the mice was monitored according to score sheets approved by LAVES with pre-defined humane endpoints. Breeding pairs of NRG mice (stock number 007799, NOD.Cg- *Rag1^tm1Mom^ IL-2rg^tm1Wjl^/SzJ*) and NSG-DKO (stock number 025216, *NOD.Cg-Prkdc^scid^ H2-K1^b-tm1Bpe^ H2-Ab1^g7-em1Mvw^ H2-D1^b-tm1Bpe^ IL-2rg^tm1Wjl^/SzJ*) were obtained from The Jackson Laboratory (JAX; Bar Harbor, ME, USA) and bred and maintained in house under pathogen-free conditions. For some experiments, NSG-DKO mice were directly purchased from The Jackson Laboratory. Briefly, 5–7-week-old mice were sub-lethally irradiated (450 cGy for NRG, 150 cGy for NSG-DKO) using a [^137^Cs] column irradiator (Gammacell 2000 Elan; Best Theratronics, Ottawa, ON, Canada), and 4 h after irradiation, 1–2 × 10^5^ human CB CD34^+^ cells were injected i.v. into the tail veins of the mice. The animals received antibiotics (Cotrim-K, Ratiopharm, Ulm, Germany) in their water two days prior to the irradiation, and they continued receiving it for 14 days post-HCT. The body weights and general health of the mice were monitored three times per week after HCT.

### 2.4. In Vivo Administration of LVs into Mice and BLI Analyses

The HuNSG-DKO mice were divided into two groups. One of the groups received LV, while the other did not (see Appendix A). In detail, 1 µg of p24 equivalent of each LV was injected i.v. into the tail vein. At week 1 post-HCT HLA-DR4/fLuc, HLA-A2.1/fLuc were administered, and at week 8 post-HCT, hGM-CSF/hIFN-α/HCMV-gB was injected. To visualize the fLuc expression, the mice were analyzed at weeks 8 and 12 post-HCT via BLI analyses using the IVIS SpectrumCT (PerkinElmer, Waltham, MA, USA) as described (Theobald et al., 2020 [11]). Briefly, the mice were anesthetized using isoflurane and shaved. Five minutes before imaging, the mice were injected i.p. with 2.5 µg of D-Luciferin potassium salt (SYNCHEM, Elk Grove Village, IL, USA) freshly reconstituted in 100 µL of PBS. Images were acquired in a field of view of C, with an f stop of 1 and medium binning for each mouse. The exposure time was kept at 300 s for each mouse. Data were analyzed using the LivingImage software version 4.5.5 (PerkinElmer, Waltham, MA, USA).

### 2.5. Blood and Tissue Collection and Processing

Immune reconstitution in the peripheral blood lymphocytes (PBLs) was monitored at 8, 12, and 20 weeks post-HCT. At the endpoint analyses (week 20 post-HCT), PBL was collected, and several tissues were biopsied (spleen, bone marrow, and thymus) and processed as previously reported [10,11]. The tissues were cryopreserved in cryo-medium (40% PBS; 50% Human Serum, Sigma-Aldrich, St. Louis, MO, USA; and 10% DMSO) and stored at −150 °C for further analysis.

### 2.6. Blood Analysis via Flow Cytometry

PBL samples were incubated with a lysis buffer (0.83% ammonium chloride/20 mM HEPES, pH 7.2) for 5 min at room temperature to remove erythrocytes. Samples were then blocked in PBS plus 10% FBS and stained with an optimum concentration of antibodies for flow cytometry (Appendix A), and additional washing was performed to remove unbound antibodies. For data acquisition, an LSR II flow cytometer (BD Biosciences, Heidelberg, Germany) was used, and analysis was performed using the FlowJo software version 10.10 (Treestar Inc., Ashland, OR, USA). The gating strategies can be seen in Appendix A. The data were visualized using the GraphPad Prism software version 9.5.0 (Dotmatics, Boston, MA, USA).

### 2.7. Tissue Analysis via Flow Cytometry

Cyropreserved single-cell suspensions were blocked with mouse IgG block (Sigma Aldrich, USA) and human FcR binding inhibitor (eBioscience, San Diego, CA, USA). Dead cells were excluded using Zombie UV dye (Biolegend, San Diego, CA, USA). Afterwards, the cells were stained for flow cytometry for 20 min at 4 °C using surface antibodies (Appendix A). Data were acquired on a Cytoflex LX flow cytometer (Beckman Coulter, Brea, CA, USA) and analyzed using the Kaluza software version 2.2 (Beckman Coulter, Brea, CA, USA). Gating strategies can be seen in Appendix A. The data were visualized using the GraphPad Prism software.

### 2.8. CyTOF

The bone marrow cells were thawed and then stained and barcoded with CD45-Cd using Stardard Biotools protocols (Appendix A). The samples were measured using HELIOS, a CyTOF system (Standard BioTools, South San Francisco, CA, USA).

Signal intensity measured in a CyTOF channel is often susceptible to interference from neighboring channels due to technological constraints. These interferences, known as spillover effects, can significantly hinder the accuracy of cell clustering. Most of the current approaches mitigate these effects through the use of additional beads for normalization, known as single-stained controls. However, this method can be costly, and it necessitates a customized panel design. To address these challenges, we employed CytoSpill [20], a tool that quantifies and compensates for spillover effects in CyTOF data without the need for single-stained controls. Mass cytometry spillover-corrected data were transformed using an inverse hyperbolic sine (arcsine) function [21] with a co-factor of 5. We further applied a 99.9% marker normalization step, at which each areasinus hyperbolicus (arcsinh)-transformed marker was normalized using its 99.9th percentile value. For cell-type identification, we carried out a two-step process. First, we performed an unsupervised clustering of cells, followed by assigning cell types to each cluster. After defining a subset of relevant markers, we utilized the FlowSOM [22] clustering algorithm, incorporating self-organizing map clustering and minimal spanning trees to cluster all cells into 100 groups based on the expression of lineage-defining markers presented in the dot plots. Subsequently, we metaclustered these initial clusters into 10 biologically relevant clusters using consensus hierarchical clustering. The final clusters underwent manual refinement and annotation based on the median expression profile of individual metaclusters. Clusters containing non-biologically meaningful signals were labeled “Unassigned”. Spillover correction was conducted utilizing the CytoSpill R package accessible on GitHub at https://github.com/KChen-lab/CytoSpill (accessed on 10 August 2024). The clustering algorithm FlowSOM (Bioconductor FlowSOM package in R) is available at https://github.com/SofieVG/FlowSOM (version 2.10.0), accessed on 10 August 2024. Plots were created in Python version 3.10 using scanpy version 1.9.3 [23], which is a scalable toolkit for analyzing single-cell data. For data visualization, high-dimensional single-cell data were reduced to two dimensions using the nonlinear dimensionality-reduction algorithm t-SNE [24]. t-SNE plots were created and visualized using scanpy. Bar plots were created using the GraphPad Prism software (version 9.5.0).

### 2.9. mRNA Sequencing and Bio-Informatic Analyses

Spleen cells that were not used for FACS analysis were resuspended in 1 mL of Trizol (Invitrogen, Waltham, MA, USA) and frozen at −80 °C until further use. The RNA was extracted using an RNeasy MinElute kit (Qiagen, Hilden, Germany). In detail, the cells frozen in Trizol were thawed on ice, and 200 µL of chloroform was added. The samples were centrifuged at 12,000× *g* for 12 min at 4 °C. Then, the aqueous phase was transferred to a new tube, and the same volume of 70% ethanol was added. The mixture was transferred to an RNeasy MinElute spin column and centrifuged at 8000× *g* for 15 s at RT. The flow-through was dumped, and 500 µL of RPE buffer was added. The samples were once again centrifuged at 8000× *g* for 15 s at room temperature, and the flow-through was dumped. Then, 500 µL of 80% ethanol was added, and the samples were centrifuged at 8000× *g* for 2 min at RT. Afterwards, the samples were dried via centrifugation at the max speed for 5 min. The RNA was extracted using 14 µL of RNase-free water and centrifugation at the max speed for 1 min. The quantity and quality of RNA were checked using a photometer (NanoPhotometer N60, Implen, Munich, Germany). NGS analyses were carried out at the production site, the Cologne Center for Genomics (CCG). Alignment was performed using the Nextflow rnaseq pipeline (version 3.12.0) [25]. For each batch of data, a DualSeq approach (measuring two separate species simultaneously) was performed by combining the human and mouse genomes and performing mapping using the ENCODE standard options for the STAR aligner [26], in addition to modifying the “outFilterMultimapNmax” parameter from 20 to 40, as we expected many potential overlapping mappings between the two fairly homologous genomes. The counts reported through the pipeline were then extracted and filtered to exclude any human genes with an average count > 5 in the mouse-only samples, and vice versa. Next, each sample type (mouse-only and humanized mice) was subjected to a batch correction approach using the Combat_seq pipeline within the sva R package (version 3.52.0) [27]. The resulting data sets were then analyzed using DESeq2 (version 1.44.0) [28]. The DESeq models were only evaluated against the two groups that were compared. Identified differentially expressed genes were then first separated into mouse and human-specific genes and then functionally annotated using the gprofiler2 package (version 0.2.3). The resulting gene ontology biological processes were then further processed using the rrvgo package (version 1.16.0) to derive semantic similarity plots for further interpretation.

### 2.10. Statistics

For the blood values, a 3-way ANOVA with the categorical variables mouse, week, and treatment group was applied in order to test whether the repeated measures reflected by the mouse variable had an influence. Since there was no significant influence of the repeated measures, pairwise *t*-tests were applied. Statistical analysis was carried out with the statistical software R version 4.2.3. All mean values with standard deviations and calculated *p*-values can be found in Appendix A for FACS and Appendix A for CyTOF data.

## 3. Results

### 3.1. Validation of Three Different Preparations of Human CD34^+^ Isolated HPCs in Sub-Lethally Irradiated NRG Reference Mice

CD34^+^ cells isolated from three different CB donors (all positive for HLA-A*02:01) were used to transplant 5–7-week-old mice after sub-lethal irradiation. NRG mice were irradiated with 450 cGy and used as our laboratory-reference immunodeficient strain to confirm CD34^+^ cell engraftment and long-term B- and T-cell immune reconstitution (Figure 1A). The animals were routinely monitored for their well-being according to score sheets with defined human endpoints that were approved by the animal welfare institutional office. The NRG mice gained weight during the experiments and did not show signs of GvHD (Appendix A), indicating no pathology concerns, such as infections or contaminating T-cells in the HPC grafts. Circulating human cells in the blood were measured via flow cytometry as frequencies at weeks 8, 12, and 20 after HCT (the antibodies used in the study are shown in Appendix A). The total numbers of human lymphocytes homing in lymphatic tissues (bone marrow, thymus, and spleen) were analyzed at week 20 after HCT (the gating strategies are shown in Appendix A). The huNRG mice showed huCD45^+^ frequencies in blood above 14% at all the measured time points (percentage of huCD45^+^ cells in blood at week 20 post-HCT: 14.9 ± 15.3; Figure 1B and Appendix A). As expected, initially, high frequencies of B-cells were detectable, while T-cells became conspicuous at week 20 post-HCT (percentage of huCD3^+^ cells within huCD45^+^ population in blood at week 20 post-HCT: 5.4 ± 2.9; Figure 1B and Appendix A). The average of CD4^+^ T-cell frequencies in the blood was always higher than that of the CD8^+^ T-cells. All lymphatic tissues contained huCD45^+^ cells (Figure 1C–E) and huCD34^+^ cells, confirming the long-term human hematopoietic repopulation (amount of huCD34^+^ cells in the bone marrow: 1,036,343 ± 921,003; Figure 1C and Appendix A). Human CD3^+^ T-cells were detectable in all tissues with similar CD4^+-^-to-CD8^+^ ratios. The total numbers of double positive (DP) CD4^+^CD8^+^ T-cells were, in general, half a log lower than the single positive (SP) CD4^+^ and CD8^+^ T-cells. The amount of huCD3^+^ cells recovered from the thymuses was quite variable (2,281,408 ± 3,945,192; Figure 1D and Appendix A), and about 10% of them were DP T-cells (amount of DP T-cells in the thymus at 20 weeks post-HCT: 230,326 ± 567,196; Figure 1D and Appendix A). Interestingly, human B-cells were abundant in the thymus (counts of huCD19^+^ cells: 1,627,855 ± 2,659,110). Thus, these results validated the HPCs’ sources as long-time repopulating in humanized mice and promoting an endogenous development of human T-cells in the thymus and expansion in different lymphatic tissues. Further, since none of the mice developed GvHD, this indicates that the T-cells indeed developed and expanded endogenously in the mice.

### 3.2. Sub-Lethally Irradiated NSG-DKO Can Be Humanized and Show Consistent Human Immune Reconstitution

CB CD34^+^ cell units validated in humanized NRG mice were also used to transplant NSG-DKO mice lacking murine MHCs. This mouse strain is radiation-sensitive, and therefore, the mice were irradiated with a lower dose, 150 cGy, as commonly used for the parental NSG strain. First, we proposed to test whether human HPCs would engraft and maintain human immune reconstitutions long-term. Second, we wanted to test whether the lack of MHCs in the host tissues would negatively affect human T-cell development. Third, we applied our previously described lentiviral (LV) i.v. delivery system for the expression of human HLAs, cytokines, and a viral antigen [17] to test whether the human immunocytes, in particular T-cells, could show signs of activation in the NSG-DKO background. Hence, we designed a combination of multicistronic LVs to deliver HLAs (LV-HLA-DR4/fLuc and LV-HLA-A2.1/fLuc) and a “vaccine” to stimulate immune responses (LV-hGM-CSF/hIFN-α/HCMV-gB). LVs expressing HLAs were administered i.v. one week post-HCT as a preconditioning step, while the LV vaccine was given eight weeks post-HCT (the experimental scheme is shown in Figure 2A, the schemes of the LV multicistronic constructs are shown in Appendix A, and the expression of DR4 and A2.1 in transduced 3T3 mouse fibroblasts is shown in Appendix A). The expression of fLuc was confirmed with BLI analyses in anatomic regions of the livers and spleens of huNSG-DKO (Figure 2B) and non-humanized NSG-DKO mice (Appendix A). The bioluminescence signal increased from week 8 to 12 post-HCT in huNSG-DKO+LV, indicating the persistent expression of the LV-encoded transgenes (Appendix A). No signs of distress were observed after LV administration, as body weights increased during the course of the experiments, indicating the well-being of animals and no signs of xeno-GvHD (Appendix A).

NSG-DKO mice could be successfully humanized, and in fact, they showed, on average, 50% huCD45^+^ cells in the blood at all measured time points, even at 8 weeks after HCT. The addition of LV delivery promoted a slight increase in the long-term human reconstitution (percentage of huCD45^+^ cells in blood at week 20 post-HCT: 57.9 ± 18.1 in huNSG-DKO and 59.4 ± 19.4 in huNSG-DKO+LV; Figure 2C and Appendix A). Initially, almost exclusively B-cells were seen for both arms, but at 20 weeks post-HCT, T-cell development could be observed as well. The percentage of huCD3^+^ cells within huCD45^+^ population in the blood at week 20 was 7.5 ± 10.7% in huNSG-DKO, and it almost doubled to 14.2 ± 18.4% in huNSG-DKO+LV (Figure 2C; Appendix A). Further, both CD4^+^ and CD8^+^ T-cells were detectable, and on average, the frequencies of SP CD4^+^ were higher than those of the SP CD8^+^ T-cells. Therefore, the analyses of blood confirmed the HPC engraftment, long-term human hematopoietic reconstitution, T-cell development, and modest but consistent responses to the LV delivery.

We then quantified the absolute counts of SP CD3^+^CD4^+^, SP CD3^+^CD8^+^, and DP CD3^+^CD4^+^CD8^+^ T-cells in lymphatic tissues (bone marrow, thymus, and spleen) (Figure 2D–F; Appendix A). Although SP CD3^+^CD4^+^ and CD3^+^CD8^+^ T-cells were clearly detectable, DP T-cells seemed to conspicuously accumulate in all lymphatic tissues, particularly in the thymus, where they represented approximately 30% of the T-cells and reached similar levels as the SP T-cells (Figure 2E; Appendix A). A trend in the higher accumulation of CD4^+^ and CD8^+^ T-cells after LV administrations compared to non-treated huNSG-DKO mice was seen in bone marrow (amount of huCD3^+^ cells in bone marrow at 20 weeks post-HCT: 203,394 ± 431,284 in huNSG-DKO vs. 752,338 ± 1,061,005 in huNSG-DKO+LV; Figure 2D and Appendix A).

In sum, despite the relevance of the epithelial MHCs for T-cell development, human SP CD3^+^CD4^+^ and CD3^+^CD8^+^ T-cells developed endogenously in several tissues of huNSG-DKO mice. Compared with our reference strain NRG, huNSG-DKO mice seemed to accumulate higher frequencies of DP CD3^+^CD4^+^CD8^+^ T-cells, which could indicate a suboptimal thymopoiesis. LV administration provided a minor enhancement of human immune reconstitution and T-cell accumulation in huNSG-DKO mice.

### 3.3. In Vivo HLA Delivery and Vaccination via LV Enhances T-Cell Activation in huNSG-DKO

In order to further assess T-cell maturation at 20 weeks post-HCT, naïve, central memory, effector memory, and terminal effector phenotypes were enumerated via flow cytometry (see the detailed gating strategy in Appendix A). In all analyzed tissues (bone marrow, thymus, and spleen), mainly effector memory T-cells were found in huNSG-DKO mice with or without LV administration (central memory and effector memory T-cells can be seen in Figure 3A–C, naïve and terminal effector in Appendix A). In the spleen, a trend of a further accumulation of effector memory T-cells was seen upon in vivo LV delivery (percentage of effector memory huCD8^+^ T-cells in spleen: 70.5 ± 24.0% in huNSG-DKO vs. 56.9 ± 21.6% in huNSG-DKO+LV; Figure 3C and Appendix A). Moreover, activation in regard to the expression levels of PD-1 and CD69 within huCD4^+^ and huCD8^+^ T-cells was analyzed (Figure 3D–F; see the detailed gating strategy in Appendix A). An increase in PD-1 expression was seen in T-cells of the thymus and spleens of mice that received LV (MFI of PD-1 in huCD8^+^ cells in spleens 20 weeks post-HCT: 33,598 ± 30,933 in huNSG-DKO vs. 75,615 ± 56,216 in huNSG-DKO+LV; Figure 3F and Appendix A). This coincided with an increase in CD69 expression, indicating higher huCD4^+^ T-cell activation in bone marrow and the spleen after LV administration (MFI of CD69 in huCD4^+^ cells in bone marrow 20 weeks post-HCT: 12,055 ± 4031 in huNSG-DKO vs. 14,995 ± 4015 in huNSG-DKO+LV; Figure 3D and Appendix A). Thus, although the effects were modest, LV administration served as in immunostimulatory agent of T-cells in huNSG-DKO mice. Regarding splenic B lymphocytes, comparative analyses of huNSG-DKO or huNSG-DKO+LV mice showed similar immunophenotypic patterns and no signs of B-cell activation (Figure 3G; huNRG reference mouse data shown in Appendix A, while IgA^+^, IgM^+^ and IgG^+^ cell amounts are shown in Appendix A). Therefore, whereas LV delivery moderately accentuated T-cell activation, this could not be observed for B-cells in huNSG-DKO mice.

### 3.4. High-Dimensional Cell-Clustering Data Confirm αβ T-Cell Development and Provide Evidence of Human NK and γδ T-Cells in hu-NSG-DKO Mice

Fluorescence-based flow cytometry is the most widely adopted method to quantify the percentages and absolute counts of human immune cells in humanized mice. Nonetheless, the numbers of markers analyzed are limited to up to twenty parameters in conventional devices, and sometimes the fluorochromes are difficult to compensate for. A more advanced method allowing a higher number of markers is cytometry via the time-of-flight (CyTOF), or mass cytometry, that utilizes antibodies labeled with heavy metal isotopes. The resulting immune cell abundances are detected using a time-of-flight mass spectrometer. We developed a method to analyze untouched cryopreserved/thawed bone marrow samples (Figure 4A). As reference material to validate the method, we used human cryopreserved CB as a positive control and bone marrow recovered from non-humanized NRG and NSG-DKO mice as negative controls (Appendix A). Due to the limited availability of cryopreserved and viable samples, we analyzed mice from a single CB donor humanization. For each mouse, 1 × 10^6^ total cells were used for labeling (antibodies used for labeling are shown in Appendix A). By employing a human CD45^+^ barcoding step, the mouse cells were excluded from further analysis. We checked 30 markers simultaneously per sample, and bioinformatics analyses were performed for high-dimensional analyses via marker expression distribution across all events. A dimensionality reduction used t-distributed stochastic neighbor embedding (t-SNE) and a clustering algorithm (FlowSOM) for visualization to identify and quantify the cellular heterogeneity among the different experimental cohorts (Figure 4B and Figure 5A). The labeling of human markers was not detectable in non-humanized NRG reference or NSG-DKO control mice (Appendix A), while ten major human cell compartments were identified in humanized mice (Figure 4B,C and Figure 5A,B). For the reference huNRG mouse samples, mainly B-cells were found in bone marrow (Figure 4B). Similar to the FACS analysis (Figure 1C), the CyTOF analysis showed a larger population of CD4^+^ than CD8^+^ T-cells (Figure 4D). Additionally, γδ T-cells were detected (Figure 4D).

B-cells were the most dominant cell population detectable via CyTOF in huNSG-DKO or huNSG-DKO+LV mice (Figure 5A). A greater trend in the detectable total amounts of CD4^+^ T-cells was seen in huNSG-DKO+LV compared with huNSG-DKO (amounts of huCD4^+^ T-cells at week 20 in bone marrow: 999 ± 1625 in huNSG-DKO vs. 3054 ± 2708 in huNSG-DKO+LV; Figure 5C and Appendix A). Further, natural killer (NK) and γδ T-cells were detectable, but some of the huNSG-DKO mice that received LV showed outliers with much higher frequencies (amount of γδ T-cells at week 20 in bone marrow: 72 ± 77 in huNSG-DKO vs. 165 ± 147 in huNSG-DKO+LV; Figure 5C and Appendix A). Further, CyTOF identified human monocytes as an abundant population, and interestingly, with increased percentages in huNSG-DKO+LV compared with huNSG-DKO (amounts of monocytes at week 20 in bone marrow: 3543 ± 459 in huNSG-DKO vs. 4717 ± 2369 in huNSG-DKO+LV; Figure 5C and Appendix A). Notably, this was paired with higher frequencies of plasmacytoid dendritic cells (pDCs) and myeloid dendritic cells (mDCs) in huNSG-DKO+LV mice (Figure 5C; Appendix A). These results indicated stimulatory effects of the LV administration in the innate immune cells of bone marrow. Moreover, analyses of cells expressing HLA-DR showed a slight increase in CD8^+^ T-cells of huNSG-DKO+LV mice (HLA-DR expression on huCD8^+^ T-cells in bone marrow: 0.180 in huNSG-DKO vs. 0.236 in huNSG-DKO+LV; Figure 5D and Appendix A).

### 3.5. Identification of Human Biomarkers of Response via mRNA Sequencing Analyses

To explore additional biomarkers, we profiled the mouse/human transcriptional gene expression in humanized mice and the effects of LV delivery using bulk mRNA sequencing (Figure 6A). Since humanized mice possess genetic material from both mice and humans, the assignment of each identifiable transcript had to be assessed for the two species. For this assignment, control samples of non-humanized mice possessing only mouse genes and huPBMCs possessing only human genes were also sequenced. Since mice and humans have high genetic similarity, it was important to identify potential genes in which the aligner would erroneously assign a read to the wrong species [29]. These genes were removed from further analyses. To visualize all of the data, a heatmap was generated of all 59,314 genes analyzed across the two species, and it showed that the majority of genes were specifically assigned to the right species (Figure 6B). Control samples such as the non-humanized mouse samples showed a generally higher expression of mouse genes as compared to the other models and an extremely low value for human genes, as no expression of these genes should be observed here. As expected, mouse genes were identified near or below the detection limit in huPBMCs and human genes expressed at higher levels (Figure 6B). Similarly, several human transcripts could be identified in mRNA obtained from humanized mouse samples compared with non-humanized mice. Unexpectedly, samples obtained from the huNRG reference strain showed a higher expression of mouse genes and a lower expression of human genes, whereas this was inversed in huNSG-DKO mice (Figure 6C).

### 3.6. Identification of Human Biomarkers and Pathways of Response to LV Delivery

Since LV delivery showed modest immunophenotypic changes in the T-cells analyzed via flow cytometry and CyTOF in huNSG-DKO, we investigated whether similar effects could be identified at the transcriptomic level. The expression of the genes involved in pathways identifying a “defense against viruses” (Figure 7A) and a “response to other organisms” (Figure 7B) was compared between huNSG-DKO with and without LV administration. Notably, mice injected with LV displayed high upregulation of several genes associated with immune responses. Among the top highest identifiable transcripts associated with interferon responses were *IFI44*, *IFI6*, and *IFIT1* (Table 1). Several transcripts encoding for enzymes were identified, most of them interferon-inducible and with antiviral activities (*DHX58*, *USP18*, and *OAS1*) (Table 1). Therefore, a strong association was observed with the interferon pathway was observed, which might be explained by the expression of IFN-α and GM-CSF via the LV used for vaccination, which may have activated this expression profile.

Furthermore, the expression of TCR-α-variable chains (Figure 7C), TCR-β-variable chains (Figure 7D), and TCR-γδ-variable chains (Figure 7E) was compared. The differences in patterns between huNSG-DKO and huNSG-DKO+LV were modest and complex, but mice administered with LV seemed to exhibit vaster TCR-variable repertoires consistently, indicating higher T-cell polyclonality.

In addition to the single transcriptome analyses per sample, the cohorts were merged and interrogated regarding general patterns of responses. Thus, genes responsible for similar functions were clustered, and genome pathways or “reactomes” were identified (Figure 8 and Figure 9). Multiple human genome pathways were identifiable and significantly upregulated in huNSG-DKO+LV in comparison with NSG-DKO, particularly the reactomes defining defense responses to other organisms and viruses (Figure 8; Table 2). Since we did not exclude the mouse mRNA analyses from the analyses, we also interrogated which murine transcripts were more abundant in huNSG-DKO+LV in comparison to huNSG-DKO mice. Intriguingly, we identified several genes associated with RNA metabolism, which we assume to be relevant in activated mouse murine hematopoietic cells (Figure 9; Table 3). Thus, deep mRNA analyses enabled the analyses of several immune pathways in huNSG-DKO, and some of them were upregulated upon LV administration.

## 4. Discussion

Humanized mouse models have advanced in the last decades from basic science towards relevant preclinical models for translational research, particularly in the immunology field. As the demand for humanized mouse models grows to evaluate the potency and safety of human-specific immune-modulatory biologics [30,31], several optimizations to enhance the engraftment and development of human cells in HIS-mice are ongoing [32]. One important novel contribution of the NSG-DKO mouse strain humanized with PBMCs was a reported reduction in the xeno-GvHD, associated with non-specific human T-cell reactivity against mouse tissues [18]. Counteracting xeno-GvHD is key to conducting long-term studies to detect and mechanistically characterize bona fide immune toxicities, such as cytokine release syndrome (CRS) [18,33,34].

We have shown that fully humanized NRG mice generated after CB CD34^+^ HPC transplantation and injected with engineered DCs could maintain endogenously developed and functional human T- and B-cells devoid of xeno-GvHD for more than 20 weeks [10,11]. Other groups used immunodeficient mice implanted with human fetal thymus and infused with autologous fetal CD34^+^ cells (BLT mice), which displayed the development of T-cells, B-cells, and monocytes [35]. BLT mice have been broadly used in testing immunotoxicity via CRS [36] and studying and developing drugs against HIV infection [37,38]. However, due to the limited availability of human fetal tissue and ethical concerns, other CD34^+^ sources such as mobilized peripheral blood or CB are presently more commonly used.

Despite these successes, the immunologic mechanism regarding how implanted human HPCs are differentiated into human T-cells in humanized mice remains unclear. It has been postulated that human T-cells would interact with murine MHCs of the thymus epithelial cells (TECs) for the positive selection of high-affinity TCRs and negative selection to eliminate T-cells with autoimmune TCRs. Indeed, it was shown that humanized mice with murine MHC class I knock-out developed fewer huCD8^+^ T-cells, and the knock-out of the murine MHC class II was associated with fewer huCD4^+^ T-cells [39,40], which could support the hypothesis that human T-cells require mouse MHCs for their development or expansion. Furthermore, work by several groups has explored matching the HLA alleles of human cells with transgenic HLAs expressed in mouse tissues. For example, transgenic HLA mice such as NOD.Cg-*Rag1^tm1Mom^ Il2rg^tm1Wjl^* Tg (HLA-DRA; HLA-DRB1*04:01) 39-2Kito/ScasJ (DRAG) and NOD.Cg-*Rag1^tm1Mom^ Il2rg^tm1Wjl^* Tg (HLA-A/H2-D/B2M) DvsTg (HLA-DRA, HLA-DRB1*04:01) 39-2Kito/J (DRAGA) were developed. HPCs partially matched to DRAG/ DRAGA mice (expressing HLA-DR4 and/or HLA-A2.1 genotypes) showed a significantly higher reconstitution of T- and B-cells in comparison to non-matched HPC controls [13,14,15,16]. Incidentally, some reports showed that class II HLA-DR4, but not class I HLA-A2 expression in NRG mice, favored thymic engraftment with human pro-T-cells [14].

Based on our current data, we could conclude that CD34^+^ CB HPCs engrafted and resulted in T-cell development in huNSG-DKO mice. Unexpectedly, we observed that the overall humanization efficacy (frequencies of huCD45^+^ cells) was higher in NSG-DKO in comparison to the reference NRG mice. We acknowledge that our study faced some limitations. Firstly, we could only obtain modest cohort sizes for each HPC graft used for the three experimental arms run in parallel. Secondly, using NSG instead of NRG mice as a reference would have been a more relevant strain background, as the NSG and NRG strains received different irradiation doses. In spite of this, Pearson et al. showed that NRG mice support similar levels of human lympho-hematopoietic cell engraftment as NSG mice irradiated using different doses [6]. Hence, we believe that the faster and higher human hematopoietic reconstitution in huNSG-DKO mice is associated with the knock-out of mouse MHCs. The mechanistic explanation of this new finding is beyond the scope of this current work and requires future investigation. We speculate that, due to the lack of mouse MHCs, there are fewer inflammatory host-versus-graft xenograft reactions during HPC engraftment, i.e., since the mouse myeloid cells are less activated to perform the phagocytosis of the human cells.

Our own original bias was that human T-cell development would have been dramatically impaired in huNSG-DKO mice due to the absence of MHC expression in the TECs or other lymphoid tissues. However, our data indicated that human T-cells could develop, at least to a certain extent, in huNSG-DKO and that they did not rely on the expression of the mouse MHC variant chains on the mouse tissues to do so. A likely explanation for the observed development of human T-cells is that HLAs expressed on differentiated human myeloid or lymphoid cells could promote the thymic or extrathymic development of the T-cells in huNSG-DKO mice. When humans undergo thymectomy, extrathymic T-cell development can occur, and therefore, tissues other than the TECs can interact with the developing human T-cells [41,42].

In addition, we cannot exclude the possibility that human endothelial cells present in the CD34^+^ HPC graft might have been transplanted or differentiated into the mice and served to promote human T-cell development. Furthermore, NSG-DKO mice still express the murine β2 microglobulin, which could form a complex with mouse CD1 and promote the differentiation of human NKT cells [43]. The frequencies of NKT cells can be determined by the expression of CD3 and NKT TCR (Vα24-Jα18 TCR); however, their frequencies in PBMCs are very low (less than 0.1%) [44]. Due to limitations in the total cell counts in our sampled mouse tissues, this detail could not be investigated in this study.

Another unexpected observation was that the T-cells in the thymus were mainly of the effector memory type and not naïve. Since mature T-cells can reenter the thymus [45], some of the T-cells detectable in the thymus may have migrated from other lymphatic tissues. In fact, since recirculating T-cells also express HLAs, they could also contribute to the positive and negative selection of bystander T-cells in a mouse strain that lacks MHCs [46,47]. Incidentally, it has been shown that memory CD8^+^ T-cells that reenter the thymus can be restricted to antigens and protect against infections [48]. Further, B-cells express MHC class I and II, serve as antigen-presenting cells, are very abundant in humanized mice, and can also be detected in the thymus. Thus, B-cells could be one interesting cellular component interacting with the developing thymocytes and activating them.

Previous to the current study, we showed that the administration of LVs into huNRG mice for the co-expression of HLA-DR4 and hGM-CSF/hIFN-α/HCMV-gB enhanced B-cell development compared to huNRG mice that did not receive the LV injection [17]. Although, in the current studies, we additionally included LV administration for HLA-A2.1 expression, we did not observe effects on the B-cell development in huNSG-DKO mice, and we observed only a modest effect on the overall immunity, which was best detected via mRNA sequencing analyses. Therefore, a plausible explanation is that the LV administration strategy in the NRG background seems to be more fit in promoting T- and B-cell development than the NSG-DKO strain. Nonetheless, one study compared a transgenic NSG mouse strain expressing positively for HLA-A*02:01 with NSG mice injected i.v. or intrathoracically with adeno-associated virus (AAV) expressing HLA-A*02:01 [49]. HLA-A*02:01 expression in the thymus was confirmed, and it was associated with increased HIS development in comparison to parental NSG mice not expressing HLA-A*02:01.

Although we observed a human T-cell presence in several lymphatic tissues of hNSG-DKO, their development was not optimal. We showed that about one-third of the thymocytes remained double positive CD4^+^CD8^+^ cells, and the LV-mediated HLA-DR4 and HLA-A2 deliveries did not substantially improve the development of SP CD4^+^ or CD8^+^ cells. This may reflect a limitation of our study in that we were unable to obtain cord blood units double positive for HLA-DR4 and HLA-A2, and thus, we used only HLA-A2 units.

Additionally, huNSG-DKO mice were vaccinated with LV-expressing hGM-CSF/hIFN-α/HCMV-gB to boost human immune responses. We observed that the detection of human leukocytes was modestly enhanced upon LV injections. After LV administrations, one of the outcomes was modestly higher T-cell levels in the spleens and bone marrow of mice in comparison to animals that were not treated with LV. We acknowledge that the LV combined gene deliveries used in our studies were complex and could have produced different modes of action, in addition to HLA-A*02:01 matching. One of the mechanisms could also be indirect, through the activation of human monocytes and dendritic cells, leading to T-cell and other immune effects or just a response to cytokines present in the LV. Indeed, using CyTOF, we were also able to detect natural killer cells and γδ T-cells in huNSG-DKO mice. These cell types are less frequent than αβ T-cells, but they are relevant participants in immune responses against viruses and tumors. In fact, the detection of higher levels of human natural killer cells in humanized mice has been associated with superior human immune responses [50]. Further, we observed monocytes and dendritic cells and an enhanced expression of CD69 in neutrophils of huNSG-DKO+LV. Since it was shown that GM-CSF can activate neutrophils and enhance CD69 expression [51], this suggests that the LVs promoted an immunization boost.

Our bulk mRNA analyses using bioinformatics tools to separate the murine and human transcripts showed a vast array of human genes upregulated upon LV delivery. Importantly, several dozens of human transcripts implicated in immune responses and metabolism were identified, which would not have been possible with FACS or CyTOF. Nonetheless, the transcriptomic data corroborated the main results observed with flow cytometry and CyTOF: a higher humanization and activation of immune responses in huNSG-DKO mice. In particular, mRNA expression analyses highlighted the functional activation of several pathways of human adaptive immunity, such as the upregulation of genes responsible for defense against viruses and other organisms and a higher expression of polyclonal TCRs. Excitingly, LV delivery showed a positive correlation with the higher expression of transcripts involved with RNA metabolism. Since the murine leukocytes are of myeloid origin in humanized mice, we speculate that this activation is relevant for innate responses in the model. We will certainly deepen our mRNA sequencing analyses in the future, including the selection of specified mouse/human cells at the single-cell level to obtain more comprehensive data sets.

As several human lymphocyte populations require species-specific cytokines to develop and grow, we anticipate that NSG-DKO (and NRG-DKO) mice expressing human cytokines will be available soon. Take as an example the currently popular NSG-SGM3 (NOD.Cg-*Prkdc^scid^ Il2rg^tm1Wjl^* Tg(CMV-IL3,CSF2,KITLG)1Eav/MloySzJ) mice, expressing human stem cell factor (SCF), granulocyte-macrophage colony-stimulating factor (GM-CSF), and interleukin (IL) 3. This strain is known, upon an injection of CD34^+^ CB cells, to develop not only human T-cells but also a broad myeloid compartment, in particular eosinophils and macrophages [52,53]. Moreover, the MISTRG strain (Balb/c;129S4- *Rag2^tm1.1Flv^ Csf1^tm1(CSF1)Flv^ Csf2/Il3^tm1.1(CSF2,IL3)Flv^ Thpo^tm1.1(TPO)Flv^ Il2rg^tm1.1Flv^* Tg(SIRPA)1Flv/J) expressing IL-3, GM-CSF, and additionally human signal regulatory protein α (SIRPA), thrombopoietin, and human homologues of the cytokine macrophage colony-stimulating factor shows high HIS engraftment efficiency and develops human myeloid, natural killer cells, and macrophages. Due to a heightened replacement of mouse hematopoietic progenitors in the bone marrow and incomplete human erythropoiesis, the mice can commonly become anemic and die [54]. Furthermore, humanized NSG-FLT3 (NOD.Cg-*Flt3^em2Mvw^ Prkdc^scid^ Il2rg^tm1Wjl^* Tg(FLT3LG)7Sz/SzJ) mice, with the human FLT3 ligand knocked in and mouse FLT3 receptor knocked out, have significantly higher levels of human monocytes, dendritic, natural killer, and T-cells in comparison to NSG mice [55]. A similar effect was also seen in different strains but with the same transgenic mutations to diminish mouse FLT3 and introduce human FLT3 [56]. Thus, NSG-SGM3 and NSG-FLT3 strains showing superior humanization and the development of different populations of the myeloid cell compartment would be promising candidates to be cross-bred with NSG-DKO mice.

Unfortunately, due to the scarcity of viable samples after the several analyses shown here, we could not complete the analyses of the TCR repertoire or perform antigen-specific functional tests. These studies remain to be performed in the future.

In summary, through the use different methods at the cellular or transcriptomic level, huNSG-DKO mice showed high humanization potential, and the absence of mouse MHCs did not hinder T-cell development. This contradicted the paradigm suggesting that human T-cells require mouse epithelial MHCs for development in humanized mouse models. This finding re-opens perspectives regarding the roles of recirculating T-cells, B-cells, or dendritic cells to participate in positive and negative T-cell selections. Ultimately, as HIS mice are becoming part of several translational programs to test or compare immunotherapies against cancer, the next experimental challenge will be to match the HLAs of the HPCs and HLAs expressed in mice and HLAs in the tumor. Towards this goal, artifacts caused by xeno-GvHD can be potentially reduced using the NSG-DKO strain. Further, once the HLA matching is optimized or completed, higher levels of antigen-specific T-cell responses can be measured. All in all, our proof-of-concept study will contribute to the future development of optimized NSG-DKO humanized mice for broad uses in basic immunology and translational research.

## Figures and Tables

**Figure 1 cells-13-01686-f001:**
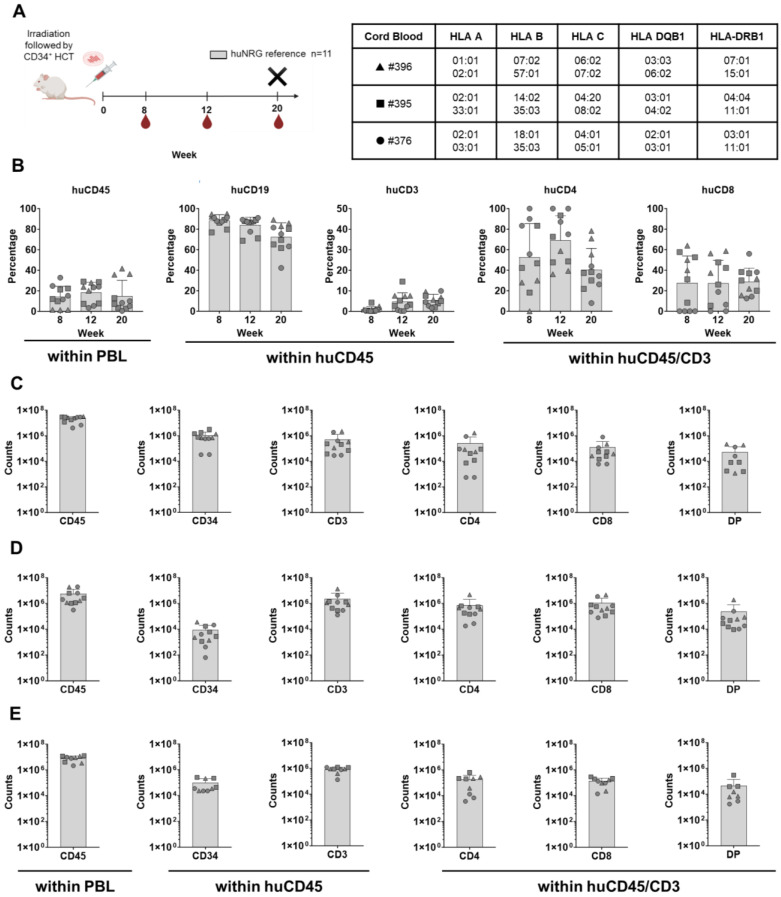
Human hematopoietic engraftment and T-cell development in humanized NRG mice used as a reference strain. (**A**) Scheme of experiments: CD34^+^ stem cell transplantation (HCT) i.v. after irradiation, blood collection (BL), and termination (X). The humanized mice were humanized with three different CB donors: #396 (depicted as a triangle), #395 (depicted as a square), and #376 (depicted as a circle). Both female and male mice were used in this experiment (details in Appendix A). Created in BioRender. Lab, S. (2024) BioRender.com/x52w302. (**B**) Blood analyses at weeks 8, 12, and 20 after HCT and longitudinal quantification of cells expressing huCD45, huCD34, huCD19, huCD3, huCD4, and huCD8 (in percentages). (**C**) Bone-marrow analyses showing the quantification of cells expressing huCD45, huCD34, huCD3, huCD4, huCD8, and double positive (DP) (in absolute cell counts, log scale). (**D**) Thymus analyses and the quantification of cells expressing huCD45, huCD34, huCD3, huCD4, huCD8, and DP (in absolute cell counts, log scale). (**E**) Spleen analyses and quantification of cells expressing huCD45, huCD34, huCD3, huCD4, huCD8, and DP (in absolute cell counts, log scale).

**Figure 2 cells-13-01686-f002:**
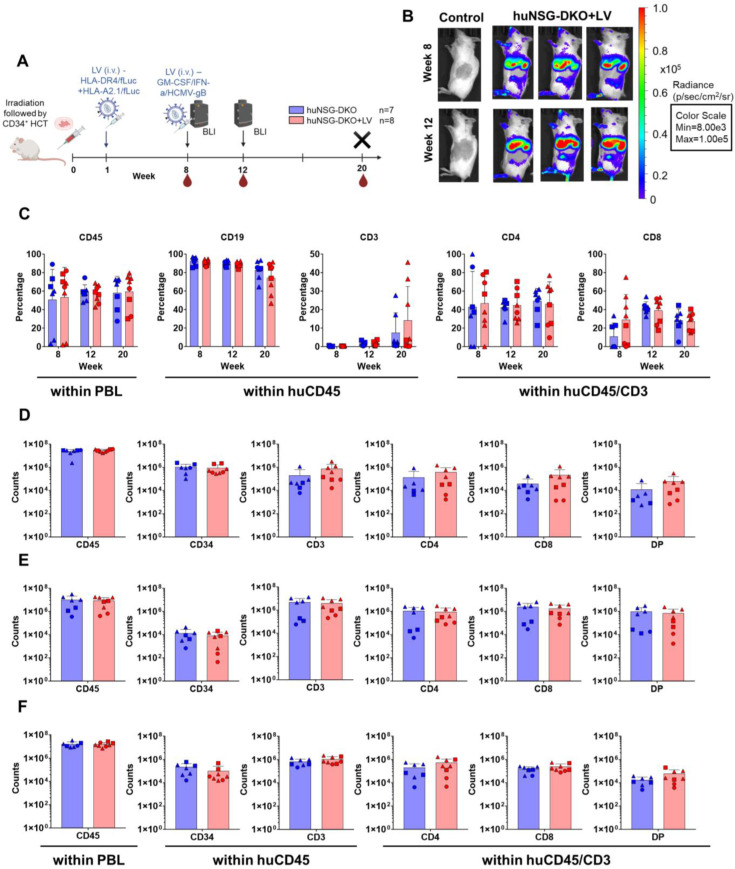
Human hematopoietic engraftment and T-cell development in humanized NSG-DKO mice. (**A**) Scheme of experiments: CD34^+^ stem-cell transplantation (HCT) i.v. after irradiation, lentivirus (LV) immunization i.v., bioluminescence imaging (BLI) analyses, blood collections (BL), and termination (X). Created in BioRender. Lab, S. (2024) BioRender.com/y83d419; (**B**) full-body BLI quantified as photons/second (p/s) at 8 or 12 weeks post-HCT of huNSG-DKO control (representative of one mouse) or after LV administration (representative of three mice). (**C**) Blood analyses at weeks 8, 12, and 20 after HCT and longitudinal quantification of cells expressing huCD45, huCD19, huCD3, huCD4, and huCD8 (in percentages). (**D**) Bone marrow analyses showing the quantification of cells expressing huCD45, huCD34, huCD3, huCD4, huCD8, and DP (in absolute cell counts, log scale). (**E**) Thymus analyses and quantification of cells expressing huCD45, huCD34, huCD3, huCD4, huCD8, and DP (in absolute cell counts, log scale). (**F**) Spleen analyses and quantification of cells expressing huCD45, huCD34, huCD3, huCD4, huCD8, and DP (in absolute cell counts, log scale).

**Figure 3 cells-13-01686-f003:**
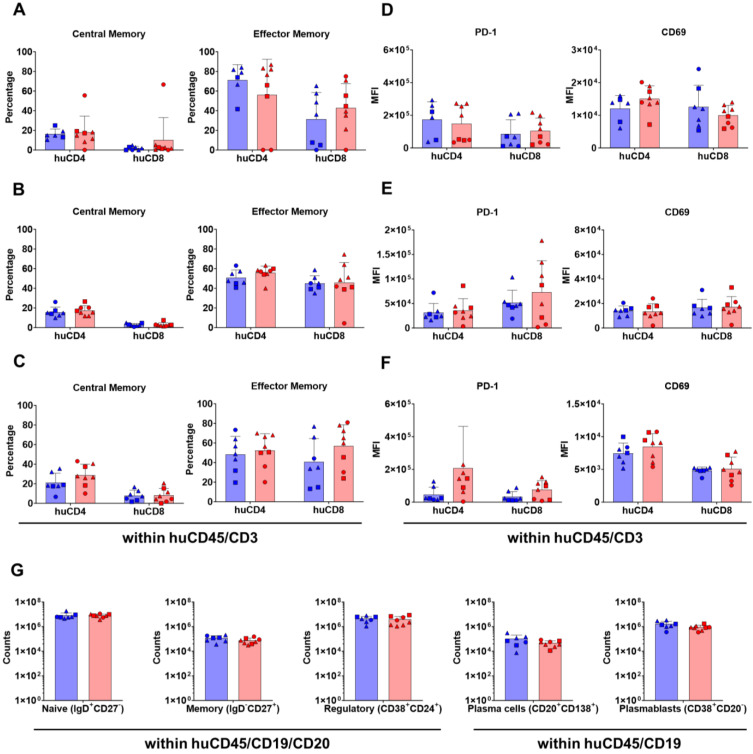
Human T-cell and B-cell maturation and activation in huNSG-DKO mice. (**A**) Analysis of central memory and effector memory T-cell subtypes within huCD4^+^ and huCD8^+^ T-cells in bone marrow (in percentages). (**B**) Analysis of central memory and effector memory T-cell subtypes within huCD4^+^ and huCD8^+^ T-cells in thymus (in percentages). (**C**) Analysis of central memory and effector memory T-cell subtypes within huCD4^+^ and huCD8^+^ T-cells in spleen (in percentages). (**D**) Analysis of T-cell activation markers PD-1 and CD69 within huCD4^+^ and huCD8^+^ T-cells in bone marrow (in absolute numbers, log scale). (**E**) Analysis of T-cell activation markers PD-1 and CD69 within huCD4^+^ and huCD8^+^ T-cells in thymus (in absolute numbers, log scale). (**F**) Analysis of T-cell activation markers PD-1 and CD69 within huCD4^+^ and huCD8^+^ T-cells in spleen (in absolute numbers, log scale). (**G**) Analysis of B-cell subtypes in spleens. B-cell subtypes: naïve, memory, regulatory, plasma cells, and plasmablasts (in absolute cell counts, log scale).

**Figure 4 cells-13-01686-f004:**
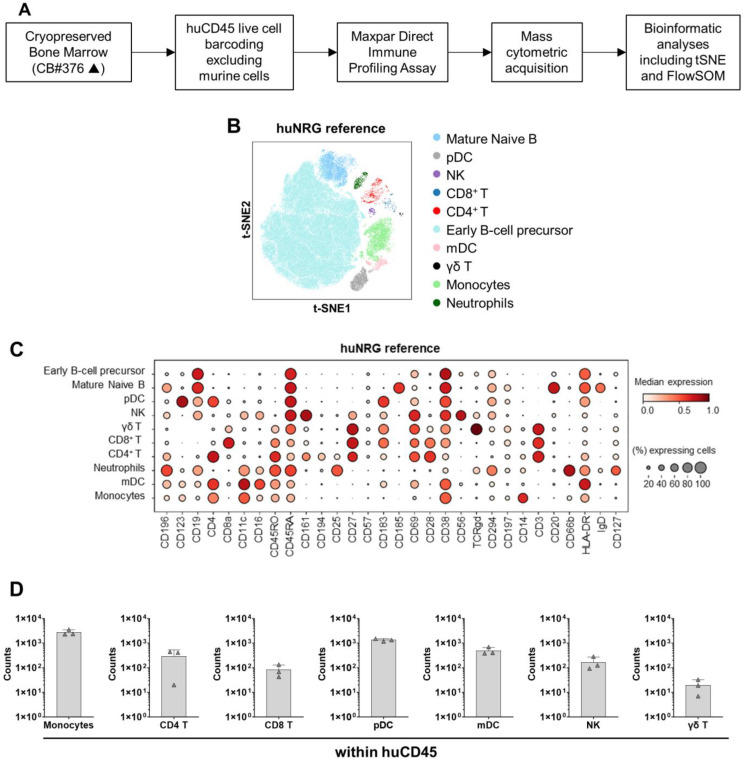
CyTOF analysis of huNRG mice as a reference. (**A**) Scheme of sample preparation, staining, CyTOF measurement, and analysis of bone marrow samples. The mice were humanized using only one donor (CB #376). (**B**) Anti-human CD45-CD live cell barcoded analysis of immune cell types in bone marrow samples 20 weeks post-HCT. A total of 152,755 cells were analyzed for huNRG mice. t-SNE plots displaying different subtypes of human immune cells clustered using FlowSOM and annotated manually using the lineage markers presented in the dot plot below. (**C**) Dotplot of huNRG cell subtypes and their expression of different Maxpar Direct Immune Profiling lineage markers. The dot size corresponds to the fraction of cells expressing the indicated marker within each cell type, and the color indicates the median expression. (**D**) CyTOF analysis of monocytes, CD4^+^, CD8^+^, and γδ T-cells, and natural killer, myeloid, and plasmacytoid dendritic cell counts in bone marrow samples (in absolute cell numbers, log scale).

**Figure 5 cells-13-01686-f005:**
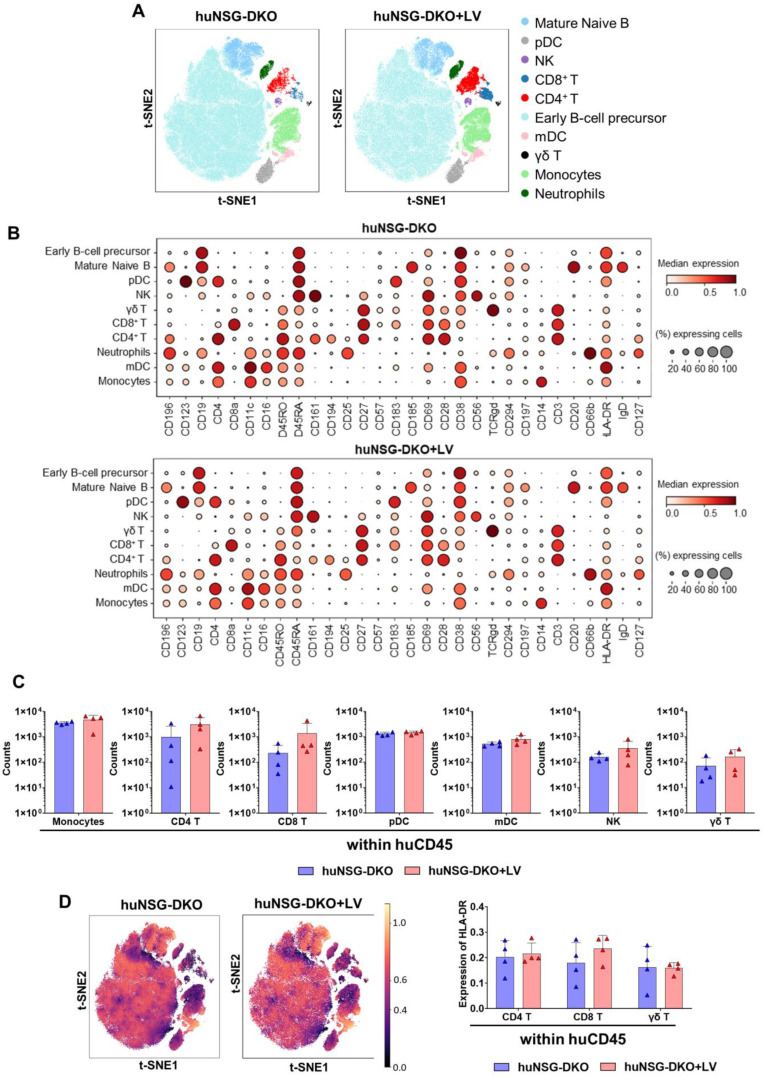
CyTOF analysis of huNSG-DKO mice. LV delivery promotes T-cell reactivity in humanized NSG-DKO mice. (**A**) Anti-human CD45-CD live cell barcoded analysis of immune cell types in bone marrow samples 20 weeks post-HCT. A total of 185,888 cells and 160,343 cells were analyzed for huNSG-DKO and huNSG-DKO+LV, respectively. t-SNE plots displaying different subtypes of human immune cells clustered using FlowSOM and annotated manually using the lineage markers presented in the dot plot (panels below). (**B**) Dotplot of huNSG-DKO and huNSG-DKO+LV cell subtypes and their expression of different Maxpar Direct Immune Profiling lineage markers. The dot size corresponds to the fraction of cells expressing the indicated marker within each cell type, and the color indicates the median expression. (**C**) CyTOF analysis of CD4^+^, CD8^+^, and γδ T-cells and natural killer, monocytes, myeloid, and plasmacytoid dendritic cell counts in bone marrow samples (in absolute cell numbers, log scale). (**D**) On right side: overlay of HLA-DR expression on t-SNE embeddings of huNSG-DKO and huNSG-DKO+LV across various cell types, as depicted in panel A. On left side: HLA-DR expression in CD4^+^, CD8^+^, and γδ T-cells (in absolute numbers).

**Figure 6 cells-13-01686-f006:**
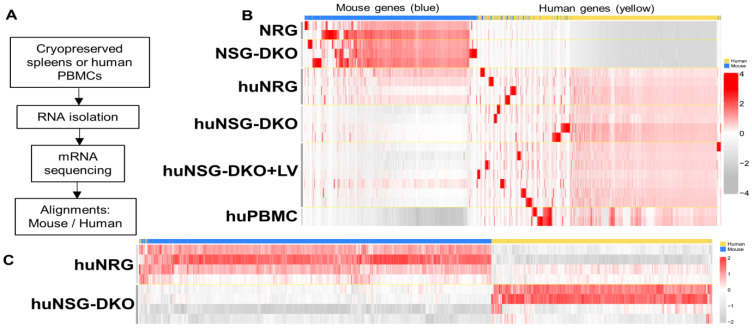
Humanized NSG-DKO mice showed better humanization than huNRG mice that were used as a reference. (**A**) Scheme of sample preparation, RNA isolation, mRNA sequencing, and analysis of spleen samples. (**B**) Alignment of mouse (blue) versus human (yellow) genes. As expected, non-humanized NRG and NSG-DKO mice have higher frequencies of mouse upregulated transcripts than humanized mice and PBMCs. HuNRG mice still upregulate some mouse genes, while in comparison, huNSG-DKO mice do not. (**C**) Differentially expressed genes between humanized NRG and NSG-DKO mice. Higher frequencies of mouse transcripts are seen in huNRG, while huNSG-DKO upregulate more human genes.

**Figure 7 cells-13-01686-f007:**
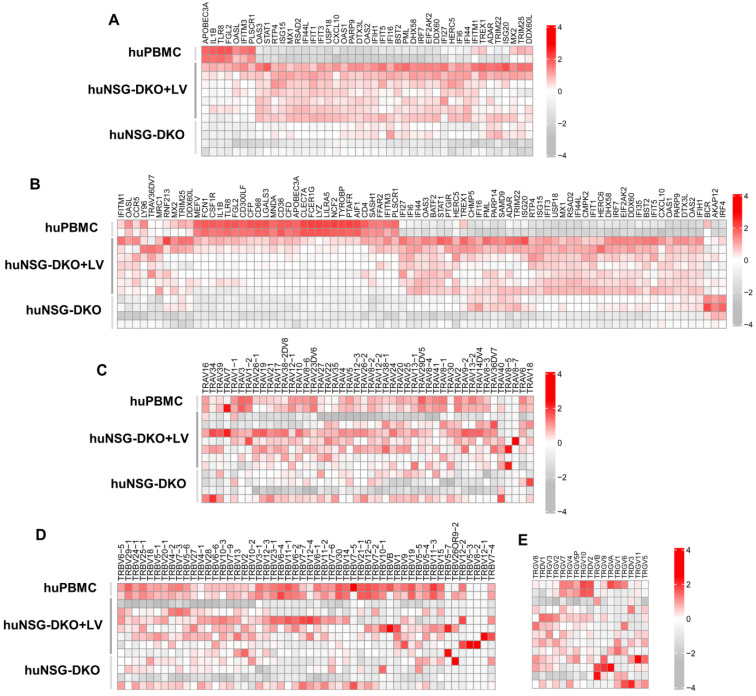
Humanized NSG-DKO mice show upregulation of several biomarkers of immune responses after LV delivery. (**A**) Genes responsible for defense response to viruses are upregulated in huNSG-DKO+LV. (**B**) Genes responsible for response to other organism are upregulated in huNSG-DKO+LV. (**C**) Recombinations in T-cell-receptor A-variable chain are polyclonal and more frequent in huNSG-DKO+LV. (**D**) Recombinations in T-cell-receptor B-variable chain are polyclonal and more frequent in huNSG-DKO+LV. (**E**) Recombinations in T-cell-receptor G- and D-variable chains are polyclonal and more frequent in huNSG-DKO+LV.

**Figure 8 cells-13-01686-f008:**
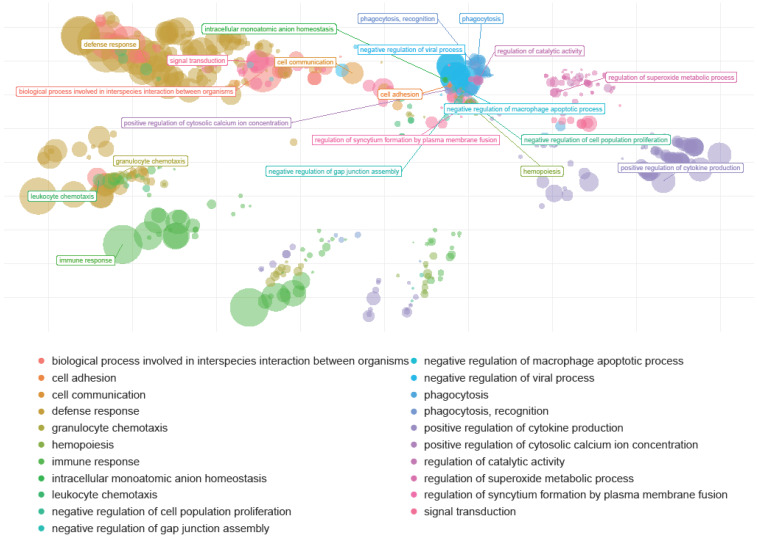
Human genome pathways upregulated in huNSG-DKO + LV in comparison to huNSG-DKO mice. Humanized NSG-DKO mice show the activation of several immune (immune response, defense response, and myeloid leukocyte activation) pathways after LV delivery. The sizes of the bubbles represent their relevance.

**Figure 9 cells-13-01686-f009:**
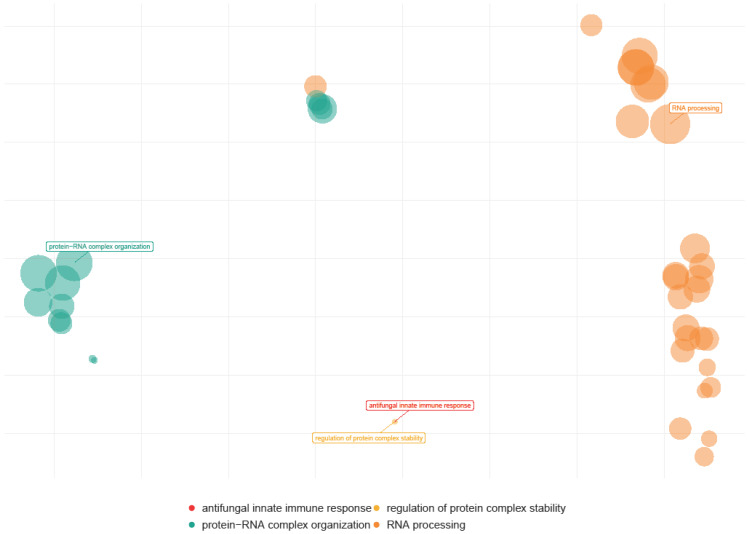
Mouse genome pathways upregulated in huNSG-DKO + LV in comparison to huNSG-DKO mice. LV delivery upregulates RNA-processing pathways within a mouse. The sizes of the bubbles represent their relevance.

**Table 1 cells-13-01686-t001:** Human biomarkers associated with TCR sequences, immune responses, and enzymes upregulated in huNSG-DKO+LV compared to huNSG-DKO.

Biomarker Class	Gene Name	Description	Term ID	p Value	Remarks
TCR Sequences	TRBJ2-5	T-cell receptor beta joining 2-5	ENSG00000211769	1.9 × 10^−14^	Predicted to be part of the TCR complex
	TRAJ33	T-cell receptor alpha joining 33	ENSG00000211856	4.2 × 10^−14^	Associated with MAIT cells
	TRAJ48	T-cell receptor alpha joining 48	ENSG00000211841	8.9 × 10^−14^	In germline T-cell receptor alpha and delta sequences
	TRAJ9	T-cell receptor alpha joining 9	ENSG00000211880	4.6 × 10^−13^	In germline T-cell receptor alpha and delta sequences
	TRGV7	T-cell receptor gamma variable 7	ENSG00000249978	1.7 × 10^−5^	Pseudogene
Immune responses	IFI44L	Interferon-induced protein 44 like	ENSG00000137959.17	1.6 × 10^−28^	Interferon-induced, antiviral, macrophage response to bacterial infection
	IFI44	Interferon-induced protein 44	ENSG00000137965.11	4.2 × 10^−26^	Interferon-induced, immune response
	IFI6	Interferon alpha inducible protein 6	ENSG00000126709.16	6 × 10^−26^	Interferon-induced, antiviral
	RSAD2	Radical S-adenosyl methionine domain containing 2	ENSG00000134321	8.8 × 10^−24^	Interferon-induced, antiviral
	IFIT1	Interferon-induced protein with tetratricopeptide repeats 1	ENSG00000185745.10	6.8 × 10^−21^	Interferon-induced, antiviral
	IFIT3	Interferon-induced protein with tetratricopeptide repeats 3	ENSG00000119917.15	9.4 × 10^−21^	Interferon-induced, antiviral
	XAF1	XIAP associated factor 1	ENSG00000132530.17	3.6 × 10^−19^	Antiviral, may be a tumor suppressor
Enzymes	DHX58	DExH-box helicase 58	ENSG00000108771.13	2.9 × 10^−21^	Negative regulation of immune response and interferon response
	USP18	Ubiquitin specific peptidase 18	ENSG00000184979.11	4.9 × 10^−20^	Interferon-induced, antiviral
	OAS1	2′-5′-oligoadenylate synthetase 1	ENSG00000089127.15	6 × 10^−19^	Interferon-induced, antiviral
	HERC5	HECT and RLD domain containing E3 ubiquitin protein ligase 5	ENSG00000138646.9	3.5 × 10^−18^	Macrophage response to bacterial infection
	OAS2	2′-5′-oligoadenylate synthetase 2	ENSG00000111335.14	1.4 × 10^−16^	Interferon-induced, antiviral
	PLSCR1	Phospholipid scramblase 1	ENSG00000188313	5.9 × 10^−16^	Interferon-induced, antiviral
	CMPK2	Cytidine/uridine monophosphate kinase 2	ENSG00000134326	2.6 × 10^−15^	Interferon-induced, antiviral

**Table 2 cells-13-01686-t002:** Human pathways upregulated in huNSG-DKO+LV compared with huNSG-DKO.

Term ID	Name	Intersection Size	*p* Value
GO:0006952	Defense response	82	9.6 × 10^−39^
GO:0043207	Response to external biotic stimulus	77	1.1 × 10^−38^
GO:0051707	Response to other organism	77	1.1 × 10^−38^
GO:0009607	Response to biotic stimulus	77	3.9 × 10^−38^
GO:0044419	Biological process involved in interspecies interaction between organisms	79	1.2 × 10^−37^
GO:0051607	Defense response to virus	42	1.7 × 10^−36^
GO:0006955	Immune response	82	3.5 × 10^−36^
GO:0098542	Defense response to other organism	67	1.0 × 10^−35^
GO:0002376	Immune system process	93	3.6 × 10^−35^
GO:0140546	Defense response to symbiont	64	4.1 × 10^−35^

**Table 3 cells-13-01686-t003:** Murine pathways upregulated in huNSG-DKO+LV compared with huNSG-DKO.

Term ID	Name	Intersection Size	*p* Value
GO:0006396	RNA processing	18	1.2 × 10^−10^
GO:0120114	Sm-like protein-family complex	12	1.0 × 10^−9^
GO:0097525	Spliceosomal snRNP complex	12	1.0 × 10^−9^
GO:0030532	Small nuclear ribonucleoprotein complex	12	1.0 × 10^−9^
GO:0022618	Protein-RNA complex assembly	12	3.7 × 10^−9^
GO:0071826	Protein-RNA complex organization	12	3.7 × 10^−9^
GO:0000398	mRNA splicing, via spliceosome	12	4.9 × 10^−9^
GO:0000377	RNA splicing, via transesterification reactions with bulged adenosine as nucleophile	12	4.9 × 10^−9^
GO:0000375	RNA splicing, via transesterification reactions	12	4.9 × 10^−9^
GO:0008380	RNA splicing	12	1.1 × 10^−8^

## Data Availability

The RNAseq data sets generated for this study can be found in the BioStudies database (http://www.ebi.ac.uk/biostudies accessed on 1 July 2024) under accession number E-MTAB-13883.

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
