# Peer review of "Long-Term Human Immune Reconstitution, T-Cell Development, and Immune Reactivity in Mice Lacking the Murine Major Histocompatibility Complex: Validation with Cellular and Gene Expression Profiles"

_cells, 2024, doi:10.3390/cells13201686_

Round 1

Reviewer 1 Report

Comments and Suggestions for Authors

The major point of this paper is that human HSCs can develop into mature peripheral T cells in mice that lack mouse MHC I and MHCII expression. This is an interesting finding to those working in the humanized mouse field. A number of points should be addressed by the authors:

1. Is it possible that stem cells exist in the CD3+CD38- population used for reconstitution that could differentiate into epithelial cells and repopulate the mouse thymus allowing differentiation of human precursors? Have the authors looked at the thymus of the reconstituted mice to determine its composition histologically?

2. As best I can tell, the mice still express mouse b2microglobulin which could combine and facilitate expression of mouse MHC like molecules such as CD1 or the neonatal Fc receptor. Could these mouse molecules be responsible for differentiation of the human cells?

3. I am confused by the authors use of the complex lentivirus containing MHC molecules, cytokine and antigen. In the end the differences seen between mice receiving the lentivirus and the controls was minimal. Could all the differences observed be secondary to the cytokines? Why is it necessary to supplement the mice with more human MHC antigens??

4. The authors detect minor population of cells via Cytof which were not detected by flow. Is this secondary to the lack of appropriate reagents for use in flow or enhanced sensitivity of the Cytof which I doubt. Please clarify. 

Comments on the Quality of English Language

A few minor correction to conform to standard English usage are needed. Not a major problem.

Author Response

  1. Is it possible that stem cells exist in the CD3+CD38- population used for reconstitution that could differentiate into epithelial cells and repopulate the mouse thymus allowing differentiation of human precursors? Have the authors looked at the thymus of the reconstituted mice to determine its composition histologically?

Answer: Thank you for the questions.

We assume that the reviewer meant the CD34+CD38- population used for reconstitution. We included a comment in the discussion addressing the possibility of human endothelial cell engraftment or differentiation:

Lines 670-672 In addition, we cannot exclude the possibility that human endothelial cells present in the CD34+ HPC graft might have been transplanted or differentiated into the mice and served to promote human T cell development.

The humanized NSG-DKO mice had small thymus structures with limited amount of cells that were all used up for flow cytometry analysis. Hence, we could not perform additional immunohistological analyses.

  1. As best I can tell, the mice still express mouse b2microglobulin which could combine and facilitate expression of mouse MHC like molecules such as CD1 or the neonatal Fc receptor. Could these mouse molecules be responsible for differentiation of the human cells?

Answer: Those are interesting possibilities. We did not observe development of mature B cells and therefore the contribution of neonatal Fc receptor may not be operating and we did not include this point in the text. We included a discussion about CD1:

Lines 672-677 Furthermore, NSG-DKO mice still express the murine β2 microglobulin, which could complex with mouse CD1 and promote differentiation of human NKT cells [43]. The frequencies of NKT cells can be determined by expression of CD3 and NKT TCR (Vα24-Jα18 TCR), however, their frequencies in PBMCs is very low (less than 0.1%) [44]. Due to limitations in the total cell counts in our sampled mouse tissues this detail could not be investigated in this study.

  1. I am confused by the authors use of the complex lentivirus containing MHC molecules, cytokine and antigen. In the end the differences seen between mice receiving the lentivirus and the controls was minimal. Could all the differences observed be secondary to the cytokines? Why is it necessary to supplement the mice with more human MHC antigens??

Answer: We apologize for the lack of clarity. We revised the introduction and discussion.

Lines 114-118 Due to the difficulties of generating transgenic mice with several different HLA alleles, in previous work we explored lentiviral vector (LV) in vivo delivery of HLA-DRB1*04:01 (DR4) into huNRG mice [17]. Upon co-administration of LV-DR4 with an additional LV ‘vaccine’ expressing human GM-CSF/IFN-α and gB, we observed activation of T and B cell development [17].

Lines 688-696 Previous to the current study, we showed that administration of LVs into huNRG mice for co-expression of HLA-DR4, hGM-CSF/hIFN-α/HCMV-gB enhanced B cell development compared to huNRG mice that did not receive the LV injection [17]. Although in the current studies we additionally included LV administration for HLA-A2.1 expression, we did not observe effects on the B cell development in huNSG-DKO mice, and only a modest effect on the overall immunity, which was best detected by mRNA sequencing analyses. Therefore, a plausible explanation is that the LV administration strategy in the NRG background seems to be fitter to promote T and B cell development than the NSG-DKO strain.

Lines 711-715 We acknowledge that the LV combined gene deliveries used in our studies was complex and could have produced different modes of action, besides the HLA-A*02:01 matching. One of the mechanisms could be also indirect, though activation of human monocytes and dendritic cells, leading to T cell and other immune effects or just a response to cytokines present in the LV.

  1. The authors detect minor population of cells via Cytof which were not detected by flow. Is this secondary to the lack of appropriate reagents for use in flow or enhanced sensitivity of the Cytof which I doubt. Please clarify. 

Answer: As the reviewer points out, indeed CyTOF has an advantage that up to 40 markers can be stained at the same time, while standard flow cytometry offers around 14 markers. This would mean that to identify same amount of populations as CyTOF one would need to do multiple flow cytometry panels, which is not always a viable option due to limited material availability. Additionally, CyTOF uses isotopes that can be easily distinguished, while flow cytometry uses fluorochromes that tend to have spillover between channels making it less sensitive to identify small cell populations.

Reviewer 2 Report

Comments and Suggestions for Authors Researchers investigated the immune responses and reactivity of humanized mice that do not express major histocompatibility complexes (MHCs) and human leukocyte antigens (HLAs). The research indicated that human immune reconstitution was observable in peripheral blood between 8 and 20 weeks following the transplantation of NSG-DKO. The development and reactivity of human T cells were not influenced by the expression of murine MHCs in humanized mice, indicating that humanized NSG-DKO presents a promising new model for investigating human immune responses. Before the introduction of clinical studies, it is crucial to contemplate and tackle these potential issues when engaging in research with this model. This study involved the isolation of CD34+ cells from three distinct cord blood donors, all of whom were positive for HLA-A*02:01. These cells were subsequently transplanted into 5-7 week old mice after sub-lethal irradiation. In numerous studies, transplant physicians regard HLA-DRB1 as more significant than HLA-A. Research indicates that alleles across various HLA loci are more significant than individual alleles, suggesting that specific combinations of alleles may indicate disease risk. Studies demonstrate that HLA-DRB1 is favored over HLA-A, -B, or -C in particular contexts. The authors need to provide a rationale for selecting HLA-A identical instead of HLA-DRB1. The HLA-DRB1 gene is primarily associated with adaptive immune responses, particularly in the regulation of T cell activation and antigen presentation. The HLA-DRB1 gene is integral to the regulation of T cell function and antigen presentation, which are vital for adaptive immune responses.

Author Response

Researchers investigated the immune responses and reactivity of humanized mice that do not express major histocompatibility complexes (MHCs) and human leukocyte antigens (HLAs). The research indicated that human immune reconstitution was observable in peripheral blood between 8 and 20 weeks following the transplantation of NSG-DKO. The development and reactivity of human T cells were not influenced by the expression of murine MHCs in humanized mice, indicating that humanized NSG-DKO presents a promising new model for investigating human immune responses. Before the introduction of clinical studies, it is crucial to contemplate and tackle these potential issues when engaging in research with this model. This study involved the isolation of CD34+ cells from three distinct cord blood donors, all of whom were positive for HLA-A*02:01. These cells were subsequently transplanted into 5-7 week old mice after sub-lethal irradiation. In numerous studies, transplant physicians regard HLA-DRB1 as more significant than HLA-A. Research indicates that alleles across various HLA loci are more significant than individual alleles, suggesting that specific combinations of alleles may indicate disease risk. Studies demonstrate that HLA-DRB1 is favored over HLA-A, -B, or -C in particular contexts. The authors need to provide a rationale for selecting HLA-A identical instead of HLA-DRB1. The HLA-DRB1 gene is primarily associated with adaptive immune responses, particularly in the regulation of T cell activation and antigen presentation. The HLA-DRB1 gene is integral to the regulation of T cell function and antigen presentation, which are vital for adaptive immune responses.

Answer: We agree with the reviewer that including a HLA-DRB1 matched system might have been more interesting. We addressed the limiations of our study in the material and methods and discussion.

Lines 165-170 Although HLA-DRB1*0401 and HLA-A*0201 should be relatively frequent in the Caucasian population [19], we were not able to obtain HLA-DRB1*0401 positive CB from our donor population for matching the LV-HLA-DR4/fLuc delivery. Therefore, we selected three cord blood units that were positive for HLA-A*0201, which could match the LV-HLA-A2.1/fLuc delivery.

Lines 705-706 This may reflect a limitation of our study in that we were unable to obtain cord blood units double positive for HLA-DR4 and HLA-A2, and thus used only HLA-A2 units.

Round 2

Reviewer 1 Report

Comments and Suggestions for Authors

NONE